# Enzymatic Biotransformation of Balloon Flower Root Saponins into Bioactive Platycodin D by Deglucosylation with *Caldicellulosiruptor bescii* β-Glucosidase

**DOI:** 10.3390/ijms20163854

**Published:** 2019-08-07

**Authors:** Tae-Geun Kil, Su-Hwan Kang, Tae-Hun Kim, Kyung-Chul Shin, Deok-Kun Oh

**Affiliations:** 1Department of Bioscience and Biotechnology, Konkuk University, Seoul 05029, Korea; 2Research Institute of Bioactive-Metabolome Network, Konkuk University, Seoul 05029, Korea

**Keywords:** *Caldicellulosiruptor bescii*, β-glucosidase, hydrolytic activity, platycodin D saponin, platycodi radix, biotransformation

## Abstract

Platycodin D (PD), a major saponin (platycoside) in Platycodi radix (balloon flower root), has higher pharmacological activity than the other major platycosides; however, its content in the plant root is only approximately 10% (*w*/*w*) and the productivities of PD by several enzymes are still too low for industrial applications. To rapidly increase the total PD content, the β-glucosidase from *Caldicellulosiruptor bescii* was used for the deglucosylation of the PD precursors platycoside E (PE) and platycodin D3 (PD3) in the root extract into PD. Under the optimized reaction conditions, the enzyme completely converted the PD precursors into PD with the highest productivity reported so far, increasing the total PD content to 48% (*w*/*w*). In the biotransformation process, the platycosides in Platycodi radix were hydrolyzed by four pathways: deapiosylated (deapi)-PE → deapi-PD3 → deapi-PD, PE → PD3 → PD, polygalacin D3 → polygalacin D, and 3″-*O*-acetyl polygalacin D3 → 3″-*O*-acetyl polygalacin D.

## 1. Introduction

*Platycodon grandiflorum*, a common vegetable commonly known as “balloon flower”, has been used as a health food to make side dishes, desserts, teas, and flavored liquors in Northeast Asia. It has also been used as a traditional herbal medicine for its antibacterial activity [1]. Over the past decade, interest in the saponins of Platycodi radix (the root of *Platycodon grandiflorum*), named platycosides, has increased owing to the new discovery of their pharmacological potential for treating geriatric diseases such as diabetes [2] and hyperlipidemia [3]. Recently, platycodin D (PD), a major platycoside in Platycodi radix, has been shown to have various pharmacological and nutraceutical activities, including anti-inflammatory [4], anti-allergy [5], anti-obesity [6], and antitumor effects [7]. PD was also found to stimulate apoptosis in skin cells [8].

Platycosides have two side sugar moieties attached to the pentacyclic triterpene aglycone. One sugar moiety (usually comprising 1–3 β-glucose molecules) is linked to C-3 in the aglycone by a glycosidic bond, whereas the other sugar moiety (an oligosaccharide residue of arabinose, rhamnose, xylose, and apiose) is bound to C-28 by an ester linkage (Figure 1) [9].

Deglycosylated saponins are better absorbed into the body and have a better biological effect than their glycosylated counterparts [9,10]. The two PD precursors, platycoside E (PE) and platycodin D3 (PD3), can be converted into PD through the deglucosylation reaction catalyzed by β-glucosidase. Several enzymes have been used for converting glycosylated platycosides in Platycodi radix extract into deglycosylated platycosides. The crude enzyme of *Aspergillus niger* [11] converted PD into deapiosylated and dexylosylated platycodin D (deapi-xyl-PD). Commercial enzymes, such as cellulase [12], snailase [13], and laminarinase [14], converted PE and deapiosylated platycoside E (deapi-PE) into PD and deapiosylated platycodin D (deapi-PD) via PD3 and deapiosylated platycodin D3 (deapi-PD3), respectively. β-Glucosidase from *Aspergillus usamii* [15] also converted PE and PD3 in Platycodi radix extract into PD. However, the productivities of PD by these enzymes are still too low for industrial applications.

In this study, the enzymatic reaction conditions, such as pH, temperature, and enzyme concentration, were optimized for the effective production of PD. Under optimized conditions, a thermostable β-glucosidase from *Caldicellulosiruptor bescii* was applied for the complete biotransformation of PE and PD3 in Platycodi radix extract into the bioactive PD.

## 2. Results and Discussion

### 2.1. Substrate Specificity of C. bescii β-Glucosidase for Platycosides

β-Glucosidase from the hyperthermophilic bacterium *C. bescii* has been used for the complete biotransformation of protopanaxadiol-type saponins into 20-*O*-β-glucopyranosyl-20(S)-protopanaxadiol (compound K) by deglycosylation because of its high hydrolytic activity. This enzyme is a non-specific enzyme that can hydrolyze not only glucose but also other monosaccharides such as galactose, arabinose, and xylose [16]. The β-glucosidase-encoding gene from *C. bescii* was cloned and expressed in *Escherichia coli* as described previously [16]. Using heat treatment, the expressed enzyme was purified as a soluble protein from the crude extract. To investigate the substrate specificity of the purified β-glucosidase for platycosides, its specific activity for PE, deapi-PE, PD3, deapi-PD3, PD, polygalacin D3, polygalacin D, and platyconic acid A was determined (Table 1). Among these substrates, the highest specific activity was observed with deapi-PE, followed by PE, polygalacin D3, PD3, and deapi-PD3, which were converted into deapi-PD3 and deapi-PD, PD3 and PD, polygalacin D, PD, and deapi-PD, respectively. However, no activity was found for PD, deapi-PD, polygalacin D, and platyconic acid A. These results indicated that this *C. bescii* β-glucosidase hydrolyzed only the outer two glucose molecules at C-3 of the platycosides but not the inner glucose at C-3 and the other sugar moiety at C-28.

### 2.2. Determination of Biotransformation Pathways of Platycosides in Platycodi Radix Extract Deglucosylated by C. bescii β-Glucosidase

The total concentration of platycosides in the 7.4% (*w*/*v*) Platycodi radix extract was 2.51 mg/mL, whereas the concentrations of PE, polygalacin D, PD, platycodin A, and 3″-*O*-acetyl polygalacin D3 (as the main compounds) were 1.00, 0.73, 0.27, 0.17, and 0.16 mg/mL, corresponding to 39.74%, 29.14%, 10.60%, 6.77%, and 6.18% (*w*/*w*) of the total platycosides, respectively (Table 2). The main compounds found in Platycodi radix extracts in other reports depended on the cultivation area and solvent extraction method used [9,17,18,19]. The minor platycosides were deapi-PE, polygalacin D3, PD3, deapi-PD, and deapi-PD3.

*C. bescii* β-glucosidase hydrolyzed the reagent-grade platycosides deapi-PE (1), PE (2), polygalacin D3 (5), and 3″-*O*-acetyl polygalacin D3 (9) at only the outer one or two glucose molecules at C-3. Thus, the enzyme-catalyzed biotransformation pathways of the platycosides in Platycodi radix were determined (Figure 2) as deapi-PE (1) → deapi-PD3 (3) → deapi-PD (6), PE (2) → PD3 (4) → PD (7) (Appendix A), polygalacin D3 (5) → polygalacin D (8) (Appendix A), and 3″-*O*-acetyl polygalacin D3 (9) → 3″-*O*-acetyl polygalacin D (11) (Appendix A). To the best of our knowledge, this is the first report on the pathways involved in the biotransformation of all platycosides in Platycodi radix through hydrolysis of the outer two glucose molecules at C-3.

### 2.3. Optimization of the pH and Temperature for Platycodin D Production by C. bescii β-Glucosidase Substrate Specificity of C. bescii β-Glucosidase for Platycosides

The hydrolytic activity of the purified β-glucosidase was measured over a pH range of 4.5–6.5 at 80 °C, using PE as the substrate. The maximum activity was observed at pH 5.5 (Appendix A), whereas the activity at pH 4.5 or 6.5 was less than 30% of the maximum activity, indicating that the enzyme activity was sensitive to pH changes. The effect of temperature on the enzyme activity was examined at pH 5.5, where the maximum activity was observed at 80 °C (Appendix A). The relative activity was 60% at 70 °C, and it dropped to almost 20% at 90 °C. The effect of temperature on enzyme stability was investigated by varying the temperature from 70 to 90 °C for diverse periods of time (Figure 3). The half-lives of *C. bescii* β-glucosidase for PE biotransformation at 70, 75, 80, 85, and 90 °C were 136, 30, 4.5, 0.07, and 0.03 h, respectively. The optimal reaction temperature was determined as 80 °C because the half-life of the enzyme at this temperature (4.5 h) was longer than the reaction time (2.3 h) for the biotransformation of platycosides in Platycodi radix.

The optimal pH and temperature of *A. usamii* β-glucosidase for the conversion of PE and PD3 into PD were pH 6.0 and 40 °C, respectively [15]. The optimal pH and temperature of the commercial enzymes cellulase [12] and snailase [13] for similar substrate conversions were pH 5.0 and 37.5 °C and pH 4.5 and 43 °C, respectively. The maximal activity of *C. bescii* β-glucosidase for ginsenoside Rb1 as a substrate occurred at pH 5.5 and 80 °C [16], which were the same pH and temperature as that used for PE as a substrate. The half-lives of the enzyme for ginsenoside Rb1 at 70, 75, 80, 85, and 90 °C were 96, 29, 6.2, 0.1, and 0.03 h, respectively [16]. Therefore, the thermal stability of *C. bescii* β-glucosidase was constant regardless of whether ginsenosides or platycosides were used as the substrate.

### 2.4. Optimization of Enzyme Concentration for Platycodin D Production by C. bescii β-Glucosidase

To investigate the effect of the enzyme concentration on the production of PD from reagent-grade PE, the concentration of β-glucosidase was varied from 0.01 to 0.2 mg/mL (Appendix A). PD production increased with increasing enzyme concentration up to 0.07 mg/mL. However, at concentrations above 0.07 mg/mL, the production of PD reached a plateau, indicating that the optimal enzyme concentration for PD production was 0.07 mg/mL. To evaluate the effect of the enzyme concentration on the production of PD from PE in Platycodi radix extract, the concentration of β-glucosidase was varied from 0.15 to 1.2 mg/mL (Appendix A). PD production increased with the increase in the enzyme concentration up to 0.5 mg/mL and then reached a plateau thereafter. Thus, the optimal enzyme concentration for PD production using Platycodi radix extract was 0.5 mg/mL.

### 2.5. Biotransformation of Reagent-grade Platycoside E and of Platycosides in Platycodi Radix into Platycodin D by C. bescii β-Glucosidase

The purified β-glucosidase at 0.07 mg/mL completely converted 1 mg/mL reagent-grade PE to 0.79 mg/mL PD via PD3 within 1.7 h, with a molar yield of 100%, a total productivity of 465 mg/L/h, and a specific productivity of 6639 mg/g/h (Figure 4a). The enzyme at 0.5 mg/mL completely converted 1 mg/mL (0.65 mM) PE and 0.04 mg/mL (0.03 mM) PD3 in the Platycodi radix extract into 0.83 mg/mL (0.68 mM) PD after 2.3 h, with a molar yield of 100%, a total productivity of 361 mg/L/h, and a specific productivity of 722 mg/g/h. As a result, the PD concentration was increased from 0.27 mg/mL to 1.10 mg/mL (Figure 4b). The reaction time for the complete biotransformation of platycosides in the Platycodi radix extract into PD was 1.5-fold longer than that for the complete biotransformation of reagent-grade PE.

The chromatogram from the HPLC analysis of Platycodi radix extract showed typical peaks for deapi-PE (1), PE (2), deapi-PD3 (3), PD3 (4), polygalacin D3 (5), deapi-PD (6), PD (7), polygalacin D (8), 3″-*O*-acetyl polygalacin D3 (9), and platycodin A (10) (Figure 5a). After 2.3 h, the platycosides in Platycodi radix extract (1)–(9) were converted into deapi-PD (6), PD (7), polygalacin D (8), and 3″-*O*-acetyl polygalacin D (11) (Figure 5b).

### 2.6. Comparison of Platycodin D Production from Platycosides in Platycodi Radix Extract by C. bescii β-Glucosidase with that by Other Enzymes

In another study, snailase at 150 mg/mL completely converted deapi-PE plus PE into 14.81 mg/mL of deapi-PD plus PD after 22 h, with a total productivity of 673 mg/L/h and specific productivity of 4.49 mg/L/h [13]. However, the production of PD by snailase was not exactly known. The specific productivity of PD by *C. bescii* β-glucosidase in this study was 160-fold higher than that of deapi-PD plus PD by snailase. The commercially available cellulase from *Trichoderma reesei* completely converted PE and PD3 in a Platycodi radix extract into 0.2 mg/mL PD after 24 h, with a total productivity of 0.85 mg/L/h [12]. The crude enzyme from *Cyberlindnera fabianii* produced 0.17 mg/mL PD from platycosides in Platycodi radix extract after 72 h, with a total productivity of 2.42 mg/L/h and a molar yield of 43% [20]. *A. usamii* β-glucosidase converted approximately 0.2 mM of PE and PD3 in Platycodi radix extract into 0.2 mM (0.24 mg/mL) PD within 2 h, with a total productivity of 122 mg/L/h and a specific productivity of 40.7 mg/g/h [15], which were the previous highest reported productivities. The total and specific productivities of PD by the *C. bescii* β-glucosidase were 3.0-fold and 18-fold higher than those of the extracellular *A. usamii* β-glucosidase (Table 3). Therefore, the β-glucosidase from *C. bescii* completely converted the platycosides in Platycodi radix extract into PD, with the highest total and specific productivities reported thus far.

## 3. Materials and Methods

### 3.1. Plasmid, Bacterial Strains, and Gene Cloning

Genomic DNA from *C. bescii* DSM 6725 (DSMZ, Braunschweig, Germany), *E. coli* ER2566 (New England Biolabs, Hertfordshire, UK), and pET-28a(+) (Takara, Shiga, Japan) were used as the DNA template source of the β-glucosidase gene (GenBank accession number ACM59590), host cells, and expression vector, respectively [16]. The β-glucosidase gene from *C. bescii* DSM 6725 was cloned using the one-step isothermal assembly method as described previously.

### 3.2. Preparation of Platycosides

The platycoside standards PE, deapi-PE, PD3, deapi-PD3, PD, polygalacin D3, and platyconic acid A were purchased from Ambo Laboratories (Daejeon, Republic of Korea). Polygalacin D was provided by Dr. Dae Young Lee of the National Institute of Horticultural and Herbal Science (Eumseong, Republic of Korea). The platycodin A (89% purity) and 3″-*O*-acetyl polygalacin D3 (90% purity) standards were purified from the Platycodi radix extract. Deapi-PD (89% purity) and 3″-*O*-acetyl polygalacin D (90% purity) were purified from the deglucosylated platycosides obtained from the Platycodi radix extract that had been biotransformed by *C. bescii* β-glucosidase. Preparation of the platycosides was carried out by preparative high-performance liquid chromatography (Prep-HPLC) (Agilent 1260; Agilent, Santa Clara, CA, USA) on an instrument equipped with a Hydrosphere C18 prep column (10 mm× 250 mm, 5 μm particle size; YMC, Kyoto, Japan), as described previously [9]. The column was eluted with water at a flow rate of 4.7 mL/min at 30 °C, and the products were detected by measurement of the absorbance at 203 nm.

### 3.3. Culture Conditions

Recombinant *E. coli* expressing the β-glucosidase gene from *C. bescii* was cultured at 37 °C in a 2 L flask containing 500 mL of Luria-Bertani medium mixed with 100 μg/mL kanamycin, with agitation at 200 rpm on a shaker. Once the optical density of the culture had reached 0.8 at 600 nm, 0.1 mM isopropyl-β-d-thiogalactopyranoside was added to induce β-glucosidase expression. The cells were then further cultured for 14 h at 16 °C, with shaking at 150 rpm.

### 3.4. Preparation of Platycodi Radix Extract

Platycodi radix was prepared using the method of ginseng root extraction as described previously [21]. One liter of 99.8% methanol was added to 100 g of dry Platycodi radix powder (purchased from a local market in Seoul, Republic of Korea), and extraction was carried out at 80 °C for 24 h. After the extraction, the solution was filtered through a 0.45 μm filter. The methanol in the filtrate was removed using an evaporator, and the residue was dissolved in 1 L of distilled water. The solution was applied to a column packed with Diaion HP-20 resin to absorb the saponins to the resin. The free sugars in the resin were eluted with distilled water. The saponins were then successively eluted with 99.8% methanol. The eluent was evaporated to remove the methanol, and the residue was dissolved in the same volume of distilled water as the original loading volume. As a result, 10% (*w*/*v*) Platycodi radix extract was produced. The concentration of this extract was further diluted to 7.4% (*w*/*v*) in order to adjust the concentrations of PE to 1 mg/mL.

### 3.5. Preparation of Enzyme

The recombinant *E. coli* cells expressing the β-glucosidase gene from *C. bescii* DSM 6725 were harvested from culture, washed with 0.85% NaCl, suspended in 50 mM citrate/phosphate buffer (pH 5.5), and disrupted by sonication on ice for 20 min. The unbroken cells and cell debris were eliminated by centrifugation at 13,000 ×*g* for 10 min at 4 °C, and the supernatant was incubated at 70 °C for 10 min. The heat-precipitated proteins were eliminated by centrifugation at 13,000 ×*g* for 10 min at 4 °C, and the supernatant was filtered using a 0.45-μm filter. The filtered solution was used as the purified enzyme.

### 3.6. β-Glucosidase Activity Assay

Unless otherwise stated, the reaction was performed at 80 °C in 50 mM citrate/phosphate buffer (pH 5.5) containing 0.05 mg/mL β-glucosidase and 0.4 mg/mL platycoside for 10 min. The specific activity of *C. bescii* β-glucosidase toward PE, deapi-PE, PD3, deapi-PD3, PD, polygalacin D3, polygalacin D, and platyconic acid A was measured by reacting various concentrations of the enzyme (0.0003–0.07 mg/mL) with 0.4 mg/mL of each platycoside for 10 min at 80 °C and pH 5.5.

### 3.7. Optimization of Reaction Conditions for Platycodin D Production

The effects of pH and temperature on the activity of *C. bescii* β-glucosidase for PE were examined by varying the pH from 4.5 to 6.5 at 80 °C, and by varying the temperature from 70 to 90 °C at a pH of 5.5, respectively. The effect of temperature on the stability of the enzyme was examined by pre-incubating the enzyme solutions at five different temperatures (70, 75, 80, 85, and 90 °C) for up to 20 h. Samples were withdrawn at regular time intervals and assayed by adding 0.05 mg/mL of the enzyme to citrate/phosphate buffer (pH 5.5) containing 0.4 mg/mL PE. The optimal enzyme concentration for the production of PD from platycosides in Platycodi radix extract was determined by varying the β-glucosidase concentration from 0.1 to 1.2 mg/mL. The reactions were carried out at 80 °C in 50 mM citrate/phosphate buffer (pH 5.5) containing β-glucosidase and 1 mg/mL PE and 0.04 mg/mL PD3 in 7.4% (*w*/*v*) Platycodi radix extract for 30 min.

### 3.8. Biotransformation of Reagent-grade Platycoside E and of Platycosides in Platycodi Radix Extract into Platycodin D

Biotransformation of reagent-grade PE into PD catalyzed by *C. bescii* β-glucosidase was carried out for 3 h at 80 °C in 50 mM citrate/phosphate buffer (pH 5.5) containing 0.07 mg/mL β-glucosidase and 1 mg/mL PE. The biotransformation of PE and PD3 in the Platycodi radix extract was carried out for 4 h at 80 °C in 50 mM citrate/phosphate buffer (pH 5.5) containing 0.5 mg/mL β-glucosidase and 7.4% (*w*/*v*) Platycodi radix extract.

### 3.9. HPLC Analysis of Platycosides

To extract platycosides from Platycodi radix for HPLC analysis, an equal volume of n-butanol was added to the Platycodi radix powder solution containing digoxin as an internal standard. The n-butanol fraction of the extracted solution was evaporated to complete dryness, and methanol was added to the residue for HPLC analysis. The platycosides were analyzed using an HPLC system (Agilent 1100) equipped with a Hydrosphere C18 column (4.6 mm× 150 mm, 5 μm particle size; YMC) and an evaporation light scattering detector. The column was eluted for 55 min at 30 °C, at a flow rate of 1 mL/min, with the following gradient of acetonitrile: water (*v*/*v*): from 10: 90 to 40: 60 for 30 min, from 40: 60 to 90: 10 for 15 min, from 90: 10 to 10: 90 for 5 min, and isocratic at 10: 90 for 10 min.

## 4. Conclusions

In this paper, β-glucosidase from *C. bescii* hydrolyzed only the two outer glucose molecules, but not the inner glucose molecule at C-3 of platycosides and the other sugar moiety at C-28. Thus, the enzyme was applied to the increase the total PD content by the deglucosylation of Platycodi radix saponins. The reaction pH, temperature, and enzyme concentration were optimized for the production of PD. Under the optimized conditions, the enzyme completely converted PE and PD3 in Platycodi radix into PD with the highest productivity reported to date, suggesting that *C. bescii* β-glucosidase is an effective enzyme in increasing the PD content rapidly. Our results may contribute to an improvement in the industrial biotransformation of the bioactive PD from Platycodi radix extract.

## Figures and Tables

**Figure 1 ijms-20-03854-f001:**
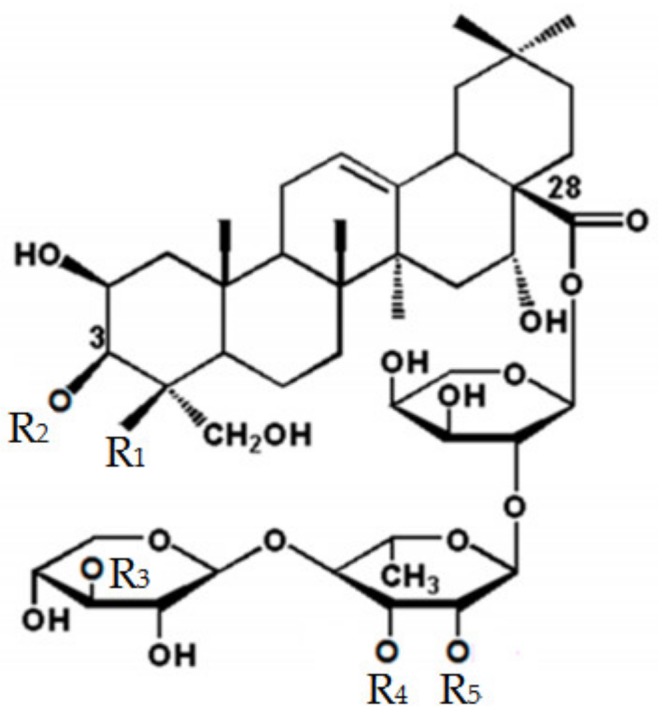
Chemical structures of triterpenoid platycosides in the roots of *Platycodon grandiflorum* and platycosides obtained after biotransformation by β-glucosidase from *Caldicellulosiruptor bescii*. Platycosides in the roots of *P. grandiflorum* are shown by numbers 1–10; the intermediate platycosides obtained during biotransformation are numbers 3 and 4; and the product platycosides obtained during biotransformation are numbers 6–8 and 11. Platycosides in the roots of *P. grandiflorum* contain glycosides at C-3 and C-28. The glycosides at C-3 are Glc, Glc-Glc, and Glc-Glc-Glc. The glycosides at C-28 are Ara-Rham (or Rham(Ac))-Xyl-Api). Glc, β-d-glucopyranosyl-; Ara, α-l-arabinopyranosyl-; Rham, α-l-rhamnopyranosyl-; Xyl, β-d-xylopyranosyl-; Api, β-d-apiofuranosyl-; and Ac, acetyl.

**Figure 2 ijms-20-03854-f002:**
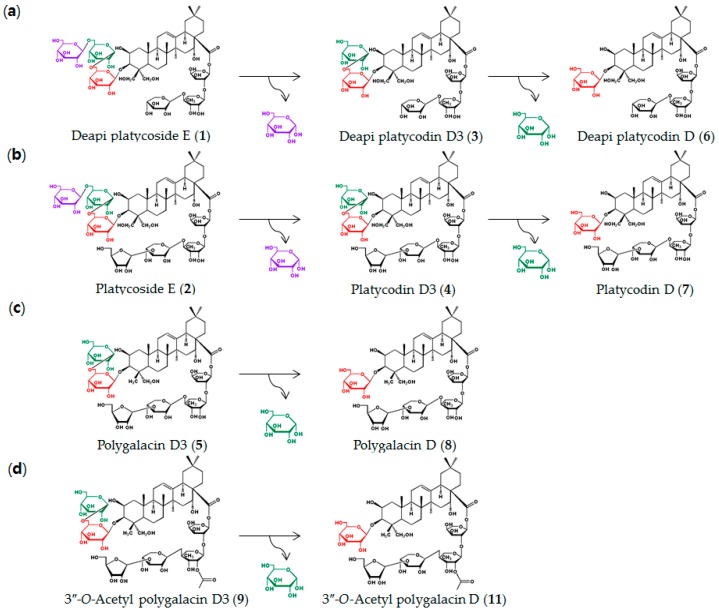
Pathways involved in the biotransformation of platycosides into deglucosylated platycosides by β-glucosidase from *Caldicellulosiruptor bescii.* (**a**) Biotransformation of deapiosylated (deapi)-platycoside E (1) into deapi-platycodin D (6) via deapi-platycodin D3 (3). (**b**) Biotransformation of platycoside E (2) into platycodin D (7) via platycodin D3 (4). (**c**) Biotransformation of polygalacin D3 (5) into polygalacin D (8). (**d**) Biotransformation of 3″-*O*-acetyl polygalacin D3 (9) into 3″-*O*-acetyl polygalacin D (11). β-Glucosidase from *Caldicellulosiruptor bescii* catalyzed the deglucosylation of only the two outer glucose molecules at C-3 of platycosides.

**Figure 3 ijms-20-03854-f003:**
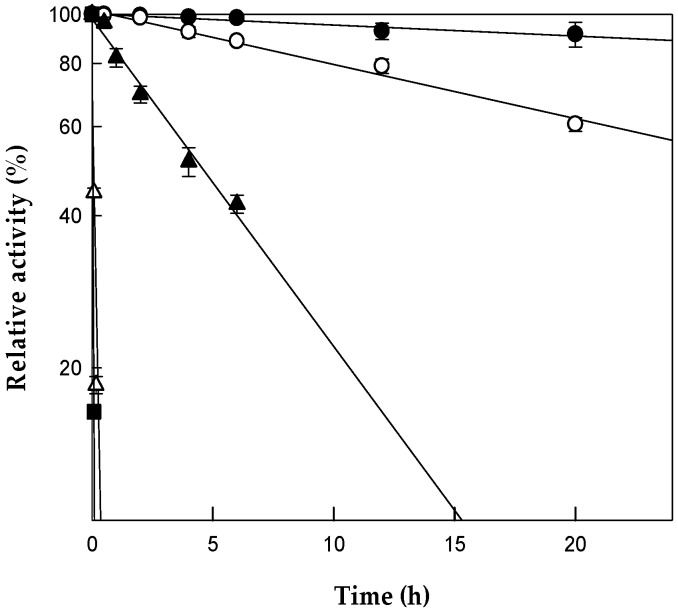
Thermal stability of β-glucosidase from *Caldicellulosiruptor bescii*. The enzyme was incubated at 70 °C (closed circle), 75 (open circle), 80 °C (closed triangle), 85 (open triangle), and 90 °C (closed square) in 50 mM citrate/phosphate buffer (pH 5.5) for various periods of time. A sample was withdrawn at the indicated time points and assayed in a reaction mixture containing 50 mM citrate/phosphate buffer (pH 5.5), 0.4 mg/mL platycoside E, and 0.05 mg/mL enzyme, at 80 °C for 10 min. Data are the means of three experiments, and error bars represent the standard deviation.

**Figure 4 ijms-20-03854-f004:**
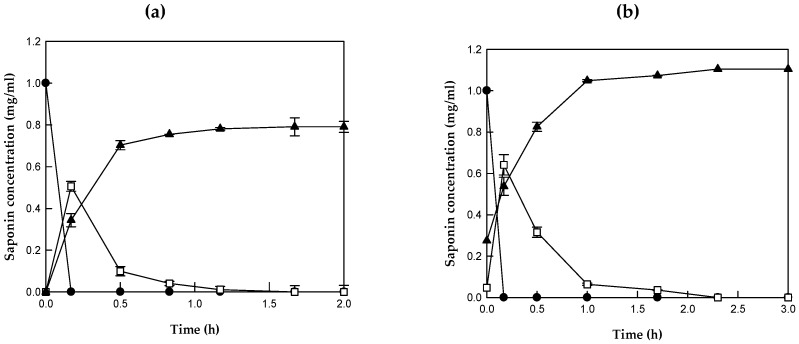
Biotransformation of (**a**) reagent-grade platycoside E (PE) into platycodin D (PD), and of (**b**) 1 mg/mL PE and 0.04 mg/mL platycodin D3 (PD3) in Platycodi radix extract into PD, by *Caldicellulosiruptor bescii* β-glucosidase under optimum reaction conditions at 80 °C. Closed circle, PE; open square, PD3; and closed triangle, PD. The PD production mixture contained 50 mM citrate/phosphate buffer (pH 5.5), 0.07 mg/mL enzyme, and 1 mg/mL reagent-grade PE; or 50 mM citrate/phosphate buffer (pH 5.5), 0.5 mg/mL enzyme, and Platycodi radix extract (containing 1 mg/mL PE, 0.04 mg/mL PD3, and 0.27 mg/mL PD). Data are the means of three experiments, and error bars represent the standard deviation.

**Figure 5 ijms-20-03854-f005:**
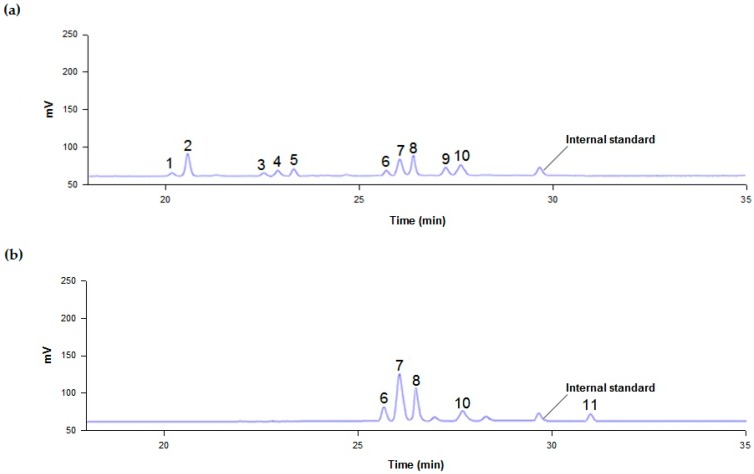
HPLC peaks of (**a**) platycosides in the Platycodi radix extract, and of (**b**) deglucosylated platycosides obtained through biotransformation by C. bescii β-glucosidase after 2.3 h. (1) Deapiosylated (deapi)-platycoside E, (2) platycoside E, (3) deapi-platycodin D3, (4) platycodin D3, (5) polygalacin D3, (6) deapi-platycodin D, (7) platycodin D, (8) polygalacin D, (9) 3″-O-acetyl polygalacin D3, (10) platycodin A, and (11) 3″-O-acetyl polygalacin D.

**Table 1 ijms-20-03854-t001:** Substrate specificity of the β-glucosidase from *Caldicellulosiruptor bescii* for platycosides.

Substrate	Product	Specific Activity (μmol/min/mg)
Deapi-PE	Deapi PD3	150.5
PE	PD3	68.3
Deapi-PD3	Deapi-PD	0.2
PD3	PD	0.4
Polygalacin D3	Polygalacin D	10.0
Deapi-PD	−	ND
PD	−	ND
Polygalacin D	−	ND
Platyconic acid A	−	ND

Deapi, deapiosylated; PD, platycodin D; PD3, platycodin D3; PE, platycoside E; ND, specific activity. not detected by the analytical methods used in this study.

**Table 2 ijms-20-03854-t002:** Content of platycosides in 7.4% (*w*/*v*) Platycodi radix extract before and after deglucosylation by *C. bescii* β-glucosidase.

No.	Platycoside	Before Reaction		After Reaction	
		Content (%, *w*/*w*)	Concentration (mg/mL)	Content (%, *w/w*)	Concentration (mg/mL)
1	Deapi-platycoside E	2.65	0.07	ND	ND
2	Platycoside E	39.74	1.00	ND	ND
3	Deapi-platycodin D3	0.29	0.01	ND	ND
4	Platycodin D3	1.77	0.04	ND	ND
5	Polygalacin D3	1.97	0.05	ND	ND
6	Deapi-platycodin D	0.88	0.02	3.53	0.08
7	Platycodin D	10.60	0.27	48.36	1.10
8	Polygalacin D	29.14	0.73	34.42	0.78
9	3″-*O*-Acetyl polygalacin D3	6.18	0.16	ND	ND
10	Platycodin A	6.77	0.17	7.51	0.17
11	3″-*O*-Acetyl polygalacin D	ND	ND	6.18	0.14
	Total	100	2.51	100	2.27

**Table 3 ijms-20-03854-t003:** Biotransformation of platycosides in the Platycodi radix extract into platycodin D.

Organism or Source	Enzyme	Total Productivity (mg/L/h)	Specific Productivity (mg/g/h)	Concentration (mg/mL)	Molar Yield (%)	Reference
*Trichoderma reesei*	Cellulase	0.85	NR	0.20	100	[12]
Snail digestive tract	Snailase	673 *^a^*	4.49 *^a^*	14.81 *^a^*	100 *^a^*	[13]
*Cyberlindnera fabianii*	Crude enzyme	2.42	NR	0.17	43	[20]
*Aspergillus usamii*	β-Glucosidase	122	40.7	0.24	99	[15]
*Caldicellulosiruptor bescii*	β-Glucosidase	361	722	1.13	100	This study

NR: specific productivity was not reported; *^a^* The values of platycodin D plus deapiosylated platycodin D.

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
