# Peer review of "Enzymatic Biotransformation of Balloon Flower Root Saponins into Bioactive Platycodin D by Deglucosylation with Caldicellulosiruptor bescii β-Glucosidase"

_ijms, 2019, doi:10.3390/ijms20163854_

Round 1
Reviewer 1 Report
The manuscript is well written, easily understandable. I have detected several point for consideration to modify:
Latin name(s) of the plant(s) are not written in italics;
In all chemical names of the compounds, heteroatoms (oxygen in this case) should be written in italics;
It should be stated in the text that the used [beta]-glucosidase is probable non-specific enzyme, because it is able to deal with other monosaccharides than glucose;
The basic criticism in this manuscript is connected with the discrepancy between Figure 1 and Figure 2. While the substituents R2 in Figure 1 are mono- to oligosaccharide units formed by glucose, no mono- or oligosaccharide moiety containing glucose is shown as R2 in Figure 2. Moreover the term deglusylation is used in the footnote of Figure 2. I believe this point needs a more detailed explanation.
I recommend to modify the manuscript accordingly and re-submit it after the corrections.
Author Response
1. Latin name(s) of the plant(s) are not written in italics.
Response) Thank you for your pointing out. The systematic name of the plant should be written in italic. As you pointed out, the word of “Platycodon grandiflorum” was revised to “Platycodon grandiflorum” in the Reference #19. (Line 379−380 of the revised manuscript)
2. In all chemical names of the compounds, heteroatoms (oxygen in this case) should be written in italics.
Response) Thank you for your suggestion. As you suggested, the word of “20-O-β-glucopyranosyl-20(S)-protopanaxadiol” was revised to “20-O-β-glucopyranosyl-20(S)-protopanaxadiol” in Results and Discussion of the section 2.1 (Line 73 of the revised manuscript). Moreover, the words of “3″-O-acetyl polygalacin D3” and “3″-O-acetyl polygalacin D” were revised to “3″-O-acetyl polygalacin D3” “3″-O-acetyl polygalacin D” (Line 24-25, 92, 99, 103, 115, 177, 178, 190, 192, 230, Figure 1, Figure 2, and Table 2 of the revised manuscript).
3. It should be stated in the text that the used [beta]-glucosidase is probable non-specific enzyme, because it is able to deal with other monosaccharides than glucose
Response) Thank you for your good comment. According to the Reference #19, β-glucosidase from Caldicellulosiruptor bescii was reported to be a non-specific enzyme for glycosides. The enzyme hydrolyzed oNP-β-D-glucopyranoside, pNP-β-D-glucopyranoside, pNP-β-D-galactopyranoside, oNP-β-D-galactopyranoside, pNP-α-L-arabinopyranoside > pNP-β-D-xylopyranoside, oNP-β-D-xylopyranoside, pNP-α-L-arabinofuranoside, and L-arabinopyranoside and L-arabinofuranoside in ginsenosides. (Reference #16 of the revised manuscript). However, β-glucosidase from Caldicellulosiruptor bescii was used to convert platycoside E into platycodin D through deglucosylation and this enzyme eliminated only two outer glucose residues of the glycosides and did not hydrolysis other sugars. As suggested, the content about non-specific enzyme was stated in the text as follows: “This enzyme is a non-specific enzyme that can hydrolyze not only glucose but also other monosaccharides such as galactose, arabinose, and xylose.” (Line 73−74 of the revised manuscript)
(16) Shin, K. C.; Kim, T. H.; Choi, J. H.; Oh, D. K., Complete Biotransformation of Protopanaxadiol-Type Ginsenosides to 20-O-β-Glucopyranosyl-20(S)-protopanaxadiol Using a Novel and Thermostable β-Glucosidase. Journal of agricultural and food chemistry. 2018, 66, (11), 2822-2829.
4. The basic criticism in this manuscript is connected with the discrepancy between Figure 1 and Figure 2. While the substituents R2 in Figure 1 are mono- to oligosaccharide units formed by glucose, no mono- or oligosaccharide moiety containing glucose is shown as R2 in Figure 2. Moreover the term deglucosylation is used in the footnote of Figure 2. I believe this point needs a more detailed.
Response) Thank you for your careful checking. To resolve the discrepancy between Figure 1 (Line 43 of the revised manuscript) and Figure 2 (Line 109 of the revised manuscript), we checked the absence of two hydrogen atoms in the hydroxyl group of L-rhamnose in Figure 1, and thus we added R4 and R5 to fill in the two hydrogen atoms that were not present. Furthermore, In Figure 2, β-glucosidase from Caldicellulosiruptor bescii was used to convert platycoside E into platycodin D, deapiosylated-platycoside E into deapiosylated-platycodin D, polygalacin D3 into polygalacin D, and 3″-O-acetyl polygalacin D3 into 3″-O-acetyl polygalacin D through deglucosylation and this enzyme eliminates only two outer glucose residues at C-3 of the glycosides. Therefore, pathways involved in the biotransformation of platycosides into deglucosylated platycosides by β-glucosidase from Caldicellulosiruptor bescii are directly related to deglucosylation. As suggested, the term deglucosylation is used in the footnote of Figure 2 as follows: “β-Glucosidase from Caldicellulosiruptor bescii catalyzed deglucosylation of only the two outer glucose molecules at C-3 of platycosides.” (Line 115−117 of the revised manuscript)

Reviewer 2 Report
The article concerns the increasing of yield of platycodin D (PD), a major saponin of Platycodi radix (balloon flower root) from the plant extract using biotransformation of other components of glycosidic fraction by extremely thermostable recombinant β-glucosidase from Caldicellulosiruptor bescii. The authors found that this enzyme eliminates only two outer glucose residues of the glycosides and doesn’t eliminated of other sugars. Such specificity allows to obtain platicodin D with 48% yield instead of 10 % without enzymatic treatment. The optimal condition (pH and temperature) was selected for enzymatic transformation. The article is well written and the results are clearly described but several errors should be corrected.
1) The chemical formulae are presented in irresponsible manner. Two hydrogen atoms are absent at hydroxyls of an L-ramnose at the Figure 1. It should be strictly corrected.
2) The formulae at the Figure 2 are presented in too low resolution. It should be fixed.
3) The formulae at the Figure 2 are wrong because glucose residue (red sugars) are drawn without CH2-6 group. It is very serious error.
4) The systematic name of the plant should be writen with italic as the name of bacterium. It should be corrected in all the corresponding places of the manuscript.
5) The name of the plant in the title of the section 2.6 “Platycodi Radix” should be replaced with “Platycodi radix”.
Hence the article is interesting and may be published but after minor but serious corrections.
Author Response
1. The chemical formulae are presented in irresponsible manner. Two hydrogen atoms are absent at hydroxyls of an L-rhamnose at the Figure 1. It should be strictly corrected.
Response) Thank you for your pointing out. To solve the problem of the absence of two hydrogen atoms in the hydroxyl group of L-rhamnose in Figure 1 (Line 43 of the revised manuscript), we newly added R4 and R5 to fill in the two hydrogen atoms that were not present.
2. The formulae at the Figure 2 are presented in too low resolution. It should be fixed.
Response) Thank you for your suggestion. Reflecting your opinion, we increased the resolution of the formula in Figure 2 (Line 109 of the revised manuscript) to make it more visible.
3. The formulae at the Figure 2 are wrong because glucose residue (red sugars) are drawn without CH2-6 group. It is very serious error.
Response) Thank you for your careful checking. It seems that there was a mistake in the process of drawing the glucose residue in Figure 2 (Line 109 of the revised manuscript). We modified Figure 2 by adding a CH2-6 group to resolve that the glucose residue was extracted without the CH2-6 group.
4. The systematic name of the plant should be written with italic as the name of bacterium. It should be corrected in all the corresponding places of the manuscript.
Response) Thank you for your good comment. As your comment, the word of “Platycodon grandiflorum” was revised to “Platycodon grandiflorum” in the Reference #19. (Line 379−380 of the revised manuscript)
5. The name of the plant in the title of the section 2.6 “Platycodi Radix” should be replaced with “Platycodi radix”.
Response) Thank you for your careful checking. As your comment, the word of “Platycodi Radix” was replaced with “Platycodi radix” in the section 2.6. (Line 194 of the revised manuscript)